# The Inflammasome-Dependent Dysfunction and Death of Retinal Ganglion Cells after Repetitive Intraocular Pressure Spikes

**DOI:** 10.3390/cells12222626

**Published:** 2023-11-15

**Authors:** Markus Spurlock, Weijun An, Galina Reshetnikova, Rong Wen, Hua Wang, Michelle Braha, Gabriela Solis, Stefan Kurtenbach, Orlando J. Galindez, Juan Pablo de Rivero Vaccari, Tsung-Han Chou, Vittorio Porciatti, Valery I. Shestopalov

**Affiliations:** 1Bascom Palmer Eye Institute Department of Ophthalmology, University of Miami Miller School of Medicine, Miami, FL 33136, USA; mspurlock@med.miami.edu (M.S.); wan@med.miami.edu (W.A.); gfr16@med.miami.edu (G.R.); rwen@med.miami.edu (R.W.); hxw628@med.miami.edu (H.W.); michellebraha@gmail.com (M.B.); gss90@miami.edu (G.S.); stefan.kurtenbach@med.miami.edu (S.K.); vporciatti@miami.edu (V.P.); 2Neuroscience Graduate Program, University of Miami Miller School of Medicine, Miami, FL 33136, USA; jderivero@med.miami.edu; 3Department of Neurological Surgery and The Miami Project to Cure Paralysis, University of Miami Miller School of Medicine, Miami, FL 33136, USA; ojg7@cornell.edu; 4Department of Cell Biology, University of Miami Miller School of Medicine, Miami, FL 33136, USA

**Keywords:** ocular hypertension, intraocular pressure spikes, retinal ganglion cells, inflammasome, caspase-1, interleukin-1β, neuroinflammation

## Abstract

The dysfunction and selective loss of retinal ganglion cells (RGCs) is a known cause of vision loss in glaucoma and other neuropathies, where ocular hypertension (OHT) is the major risk factor. We investigated the impact of transient non-ischemic OHT spikes (spOHT) on RGC function and viability in vivo to identify cellular pathways linking low-grade repetitive mechanical stress to RGC pathology. We found that repetitive spOHT had an unexpectedly high impact on intraocular homeostasis and RGC viability, while exposure to steady OHT (stOHT) of a similar intensity and duration failed to induce pathology. The repetitive spOHT induced the rapid activation of the inflammasome, marked by the upregulation of NLRP1, NLRP3, AIM2, caspases -1, -3/7, -8, and Gasdermin D (GSDMD), and the release of interleukin-1β (IL-1β) and other cytokines into the vitreous. Similar effects were also detected after 5 weeks of exposure to chronic OHT in an induced glaucoma model. The onset of these immune responses in both spOHT and glaucoma models preceded a 50% deficit in pattern electroretinogram (PERG) amplitude and a significant loss of RGCs 7 days post-injury. The inactivation of inflammasome complexes in *Nlrp1*^−/−^, *Casp1*^−/−^, and *GsdmD*^−/−^ knockout animals significantly suppressed the spOHT-induced inflammatory response and protected RGCs. Our results demonstrate that mechanical stress produced by acute repetitive spOHT or chronic OHT is mechanistically linked to inflammasome activation, which leads to RGC dysfunction and death.

## 1. Introduction

Eyes that are challenged with chronic OHT, like in the ocular hypertension type of glaucoma, are exposed to multiple mechanical, ischemic, and metabolic stresses that affect essential RGC functions, including axonal transport, energy homeostasis, electrophysiological outputs, and glial activation [1,2,3,4,5]. Repetitive short-term OHT episodes, caused by intraocular pressure (IOP) “spikes”, are also recognized as important risk factors [6,7], though these are much less studied than chronic OHT. Mounting clinical and experimental evidence from animal studies indicates that transient IOP spikes are more damaging to vision when they are recurrent and/or reach the ischemic level, such as with eye rubbing [8,9]. Recurrent OHT spikes are typically induced by medications, eye surgeries, intraocular drug injections, head-down postures during post-surgical recovery, or some lifestyle activities, including playing wind instruments, weightlifting, or head-down positions in yoga [10,11,12,13]. Repeated incidences of OHT spikes have been linked to nerve fiber layer thinning and vision loss in human eyes [8,14]. For example, repetitive intraocular drug injections were associated with vascular hypoperfusion [15,16,17] and nerve fiber layer thinning [17,18], and it has also been suggested to play a role in normal tension glaucoma [9,13,19]. Although spOHT and chronic OHT modalities differ dramatically based on the type of the initial injury, both could produce similar RGC and axon injuries that clinically present as glaucomatous [8,14]. To the best of our knowledge, the present work is the first to compare neuroinflammatory responses and their link to RGC pathology following spOHT and chronic OHT.

Neuroinflammation and the accumulation of pro-inflammatory cytokines, the two common end products of active inflammasomes [20,21,22,23], have been detected in animal [24] and human [25,26,27,28] eyes with glaucoma, along with an increase in extracellular ATP, a common inducer of the inflammasome [29]. At the cellular level, neuroinflammatory responses mediated by the release of IL-1b and tumor necrosis factor (TNFa) have been confirmed in astrocytes, Muller glia, microglia, and in RGCs in glaucomatous eyes [25,30,31,32]. Recent reports from our lab and others have shown the very early activation of neuronal NLRP1/3 inflammasome and caspase-1 (CASP1) convertase, as well as the release of IL-1β, in a rodent model of ischemic OHT injury [33,34,35], suggesting that inflammasome signaling may serve as the key trigger of glial responses. The key role of inflammation and inflammatory caspases, like caspase-8 (CASP8), in the pathogenesis of glaucoma induced via non-ischemic OHT injury has been recently shown [36,37].

An active role of inflammasome activation in the pathogenesis of retinal OHT damage [38,39] could shift the current paradigm, where inflammation is considered secondary to optic nerve damage and RGC pathology [40]. To gain insight into how short-term repetitive non-ischemic OHT spikes lead to RGC injury, we investigated the impact of the activation of retinal inflammasomes on RGC function and viability using a mouse model of spike-induced ocular hypertension (spOHT) injury. Here we report that repetitive episodes of non-ischemic IOP spikes lead to impaired RGC electrophysiological function and cell death. Our results strongly suggest that the effects of spiking OHT are mediated predominantly by acute innate immune responses, triggered by the activation of the endogenous neuronal NLRP1 inflammasome. Our results also identify key components of the inflammasome, along with its upstream regulator the mechanosensitive Panx1 channels, as therapeutic targets for OHT-mediated RGC degeneration. 

The present work is an important step towards a better understanding of the underlying molecular mechanism through spiking OHT-induced degenerative stress in RGCs. Furthermore, our results from multiple inflammasome knockout mice indicate that the acute activation of inflammasomes and their downstream products are mechanistically involved in OHT-induced RGC dysfunction and loss, while their inactivation protected RGCs.

## 2. Materials and Methods

### 2.1. Animals 

Animal handling, anesthetic procedures, experiments, and post-surgical care were performed in compliance with the NIH Guide for the Care and Use of Laboratory Animals and according to the University of Miami Institutional Animal Care and Use Committee approved protocols #18-025 and #21-036. Wildtype (WT, C57BL/6J) and transgenic mice, including *Casp1*^−/−^ (*Casp1/Casp4*(11)^del^ strain B6N.129S2-*Casp1*tm1Flv/J, Jax # 016621); *Nlrp1b*^−/−^ (B6.129S6-*Nlrp1b^tm1Bhk^*/J, Jax # 021301), and *Gsdmd*^−/−^ (C57BL/6N-Gsdmdem4Fcw/J, Jax # 032410), were obtained from Laboratories Depository (Bar Harbor, ME). The *Panx1*^−/−^ *Casp4*(11)^+/+^ (C57BL/6 background) were developed by Dr. V. Dixit and obtained from Genentech, Inc. Mice expressing ASC-citrine for ASC speck visualization were provided by Dr. D. Golenbock [41], University of Massachusetts Med. School. All animals were bred in the University of Miami animal facility and housed under standard conditions of temperature and humidity, with a 12 h light/dark cycle and free access to food and water. Equal numbers of male and female mice were assigned to each grouping.

### 2.2. Reagents

Antibodies were purchased from commercial sources: anti-GFAP (Dako, cat# z0334), anti-AIF1/Iba1 (Wako, cat# 019-19741), anti-CASP1 (Novus Biologicals, cat# IMG5028); anti-CASP1 p-20 (Adipogen, cat# AG-20B-0042); anti-IL-1β (Cell Signaling, cat# 8689S), anti-NLRP1 (Novus cat# NB100-561148SS); anti-NLRP3 (R&D, cat# AF7010); anti-RBPMS (GeneTex, cat# 118619); anti-BRN3a (SantaCruz, cat# sc31984); anti- Class III β-Tubulin (clone TUJ1, Covance); anti-GSDMDC1 (A7, Santa Cruz, cat# sc-271054); anti-CASP8 monoclonal antibody (1G12) (Enzo, cat# ALX-804-447); anti-ASC-1 (F-9, Santa Cruz, cat# sc-271054); anti-CD11b (Santa Cruz, cat# sc-271050); and anti-CD45 (Santa Cruz, cat# sc-271024). FAM-FLICA® Caspase-1 660(YVAD), FAM-FLICA® Caspase-8 550(LETD) and FAM-FLICA® Caspase-3/7 660(DEVD) were purchased from Immunochemistry Technologies, Inc. (Davis, CA, USA). 

### 2.3. The Spiking OHT Injury Model

The mean arterial pressure in murine eyes is 112 mm Hg [42], and transient non-ischemic IOP elevations to 30–40 mm Hg were shown to be non-injurious [43,44,45]. The spiking OHT model, however, was administered through seven consecutive IOP elevations of 40 mm Hg. During the procedure, the animals were under isoflurane (3%) gas anesthesia, and topical analgesia was induced with 0.5% proparacaine HCl (Bausch & Lomb Pharmaceuticals, Rochester, NY, USA). Their pupils were dilated with 1% tropicamide and 2.5% phenylephrine hydrochloride (NutraMax Products, Inc., Gloucester, MA, USA) to aid in the placement of the pressure input needle. The IOP was elevated via the cannulation of the anterior chamber with a 29G needle connected to a reservoir of normal saline (0.9% NaCl) that was placed 54 cm above the eye level to achieve a H_2_O pressure equivalent to 40 mm Hg, as described previously [33,43]. An in-line digital mini-pressure gauge (Centurion Compass CUHG; precision ± 1.5 mm Hg) was used to monitor IOP elevation and the potential loss of pressure from leaks. The spOHT experimental paradigm, shown in Figure 1A, consisted of seven consecutive 1 min IOP spikes (spOHT) achieved through a rapid rise and consecutive lowering of the reservoir, with a 1 min interval of normotension between spikes and quick transitions between normal and elevated IOP. In the steady OHT group, the IOP was elevated with a gradual 1 min increase to 40 mm Hg, which was maintained for 7 min, then returned to baseline gradually over 1 min. Thus, both models produced the same maximum IOP exposure to 40 mm Hg for a total of 7 min and were induced via a single needle insertion through the peripheral cornea. The sham control procedure was performed through the single cannulation of the anterior chamber with no elevation of the reservoir (i.e., no IOP elevation) under anesthesia for the same duration as the experimental groups.

### 2.4. The Y437H-MYOC-Induced OHT Model

In this model, IOP was elevated by over-expressing the pathogenic Y437H variant of human myocilin, as described earlier by Grotegut and Kuehn [46,47]. To over-express Y437H human myocilin, the anterior chamber of each eye of an animal was injected with 1.4 µL of Ad5-*MYOC* suspension (5 × 10^7^ pfu/eye) using a 33G needle. A gradual IOP increase to 25–35 mm was observed at 6–8 weeks in all mouse lines tested. The eyes were harvested and inner retinas were collected at 3, 5, and 8 weeks after vector injection.

### 2.5. The Retinal Ischemia–Reperfusion Model

Retinal ischemia was achieved by increasing the IOP above systolic blood pressure to 110 mm Hg for 45 min via the direct cannulation of the anterior chamber with a 29G needle connected to a normal saline (0.9% NaCl)-filled reservoir, placed at 150 cm above the eye, to create a pressure of 150 cm H_2_O (equivalent to 110 mm Hg), as previously described [33]. The pressure changes in the tube connected to the needle were calibrated using an in-line Centurion Compass CUHG1 digital pressure transducer (Centurion Medical Products Inc., Williamston, MI, USA) prior to experiments. The contralateral eyes, cannulated at normal IOP, served as normotensive controls. Complete retinal ischemia was confirmed as the whitening of the anterior segment and the blanching of the retinal arteries. The eyes were harvested 24 h post-injury; the mice were euthanized via CO_2_ overdose and the eyes were collected. The vitreous body was harvested for cytokine analysis via an ELISA (Protein Simple, Bio-Techne Inc., Minneapolis, MN, USA), and the retinas were dissected out, fixed, and processed.

### 2.6. Intravitreal Cytokine Activity Assay

To collect the vitreous body, the mice were perfused with phosphate-buffered saline (PBS), and their eyes were collected, placed on ice, and immediately dissected. The vitreous fluid was collected with three consecutive flushes of the vitreous cavity with 20 µL of sterile PBS containing a protease inhibitor cocktail. All flush samples were combined, spun for 5 min in a refrigerated centrifuge, and stored at −80 °C. ELISA kits for mouse IL-1β (R&D ID# MBL00C) or Ella SimplePlex for IL1β, TNFα, and MCP1/CCL2 (Protein Simple, Bio-Techne Inc., Minneapolis, MN, USA), as described in [48], were used to measure the cytokines released into the vitreous. Sample aliquots were processed for protein analyses in parallel with the standards and controls following the manufacturer’s instructions. A colorimetric assay was carried out using a FLUOstar Omega plate reader (BMG Labtech, Ortenberg, Germany) and analyzed using the MARS data analysis software (BMG Labtech, Ortenberg, Germany). The values from the wells containing blank samples were subtracted from the background. To validate the significance of measurements at the lowest reading, the limit of detection (LOD) and limit of quantification (LOQ) ratios were calculated from empirical data obtained in the “zero” wells of each plate, as described [49]. A minimum of three (N = 3) biological repeats were used for each data point. Significance was calculated using one-way analyses of variance (ANOVAs) followed by Tukey’s test for multiple comparisons. To measure the co-release of IL-1β, TNFα, and CCL2 cytokines, we used the Ella Simple Plex microfluidic technology (Protein Simple, Bio-Techne Inc., Minneapolis, MN, USA) with internal calibration, as described in [48].

### 2.7. In Vivo Retinal Electrophysiology and Data Analysis

An optimized protocol for PERG (pattern electroretinogram) recording in mice was previously described [50,51]. Briefly, the animals were anesthetized with ketamine/xylazine (80/10 mg/kg) and gently restrained in an animal holder. PERG signals were recorded simultaneously from both eyes from subdermal electrodes in the snout in response to horizontal bars that maximized the PERG amplitude and minimized the noise (spatial frequency, 0.05 cycles/deg; temporal frequency, 1 Hz; contrast, 100%; robust averaging of 2232 sweeps). The PERG signal-to-noise ratio was of the order of 10, and the test–retest variability was of the order of 30% [52]. Balanced salt solution (BSS) drops were applied to maintain cornea hydration. 

### 2.8. RGC Loss Assessment

To assess RGC loss, retinas were collected at 7, 14, and 21 days post-injury, fixed in 4% paraformaldehyde, and flat-mounted. RGCs were identified via RBPMS (RNA binding protein with multiple splicing) immunolabeling, visualized through confocal microscopy. RBPMS-positive cells were counted with the ImageJ plugin open-source software after thresholding and the manual exclusion of artifacts. Each retina was sampled from 16 fields in 4 retinal quadrants in 3 regions of the same eccentricities (0.5 mm, 1.0 mm, and 1.5 mm from the optic disk) as previously described [33]. RGC loss was calculated as a percentage of RBPMS-positive cells in the experimental eyes relative to the sham-operated contralateral control eyes. The cell density data (*n* ≥ 5) were averaged for each group/genotype; the data were analyzed for statistical significance with a one-way ANOVA followed by the Tukey test for multiple comparisons; and *p* values ≤ 0.05 were considered statistically significant.

### 2.9. Inflammasome Detection

The inflammasome complex formation was detected using the citrine-labeled ASC speck complex in vivo has been described previously [34]. ASC-citrine was previously shown to incorporate into oligomerizing inflammasome complexes, thus providing a surrogate inflammasome activation in mouse tissues [53]. The bioindicator mice, expressing ASC fusion protein with a C-terminal citrine protein (fluorescent GFP isoform) that brightly labels the filamentous ASC specks in vivo and allows for visualization in vivo, were provided by Dr. D. Golenbock (University of Massachusetts, MA, USA). 

### 2.10. Real-Time PCR

Gene expression was assessed via real-time PCR using gene-specific primer pairs (primer pairs were validated to span an intron and to amplify only one product (see Appendix A for details). Total RNA from 2 to 4 pooled retinas (500 ng) was extracted using Trizol and their quality was controlled using a Nanodrop. cDNA was synthesized with the Reverse Transcription System (Promega, Fitchburg, WI, USA). Real-time PCR was performed in the Rotor-Gene 6000 Cycler (Corbett Research, Mortlake, Australia) using the SYBR GREEN PCR MasterMix (Qiagen, Valencia, CA, USA); for each pair, the predicted product length was controlled via PCR. The relative expression was calculated through comparisons with a standard curve following normalization to the β-actin genes. 

### 2.11. Immunohistochemistry

The eyes were enucleated, fixed in 4% paraformaldehyde for 1 h, and cryoprotected with 30% sucrose. The retinas were embedded into the OCT media and frozen-sectioned to a thickness of 10 µm on a microtome (Leica). Slides were washed in PBS, permeabilized in PBS with 0.2% Tween20, and incubated with a primary antibody for 4–16 h. A full list of the primary antibodies used is included in the Appendix A. Retinal flat-mounts were incubated with primary antibodies for 3–5 days at 4 °C to ensure even staining. To identify target proteins, specific antibodies were diluted and incubated for 4–16 h. After washes with PBS-Tween 20, secondary antibodies were applied for 2–4 h for frozen sections and for 16 h for whole mounts. Secondary AlexaFluor dye-labeled antibodies (Thermo Fisher Scientific, Waltham, MA, USA) were applied for imaging with the Leica TSL AOBS SP5 confocal microscope (Leica Microsystems, Wetzlar, Germany); controls with primary antibodies omitted were used for specificity tests. For cell counts, single-layer scans at the midpoint between the top and bottom of the RGC layer were obtained at each sample site (8 peripheral and 6 central).

### 2.12. Statistical Analysis

Statistical comparisons of the PERG data were made using non-parametric Mann–Whitney test and Kruskal–Wallis test followed by post hoc Dunn’s multiple comparisons. The protein assay (ELISA, SimplePlex) data were presented as the mean ± standard deviation (SD) or standard error (SEM) for the RGC survival data. The real-time PCR data were analyzed with a one-way ANOVA followed by the Tukey test for multiple comparisons. For single comparisons, Student’s *t*-test was applied; the one-way ANOVA was used for between-group comparisons. GraphPad Prism software (version 6.07; GraphPad Software, La Jolla, CA, USA) was used for statistical analysis. A minimum of three biological repeats per treatment was used for in vivo IL-1β release assessment and for gene expression analysis via quantitative RT-PCR. Groups of data were compared using ANOVAs or two-tailed unpaired Student’s *t*-tests. The cell density data were analyzed with one-way ANOVAs followed by Tukey’s test for multiple comparisons. For two group comparisons, Student’s *t*-test was carried out. *p* values < 0.05 were considered statistically significant for all analyses.

## 3. Results

### 3.1. Acute Inflammasome Induction after spOHT

To characterize the effects of spOHT challenge on the inflammasome pathway, we measured the changes in transcript levels of the genes encoding for IL-1β, TNFα, caspases -1, -3, and -8, gasdermin-D (GSDMD), and the NLRP1b and NLRP3 inflammasome sensors in the sham-operated vs. experimental retinas. Significant increases in mRNA for *Il1b*, *Tnfa*, *Casp1*, *Casp8*, and *Nlrp3* genes (*n* = 3, *p* < 0.05) were detected in WT retinas after spOHT (red bars, Figure 1B), while the expression of the *Nlrp1*, *Casp3*, and *Gsdmd* genes remained unchanged. In contrast, no significant changes in the expression of these genes were detected in retinas challenged with stOHT (blue and gray bars, Figure 1B). 

Inflammasome activation in the retina was previously shown to be induced within hours after ischemic-level IOP elevation [27,33,34,35]. To determine the time course of inflammasome activation in the spOHT model, we measured IL-1β levels in the vitreous body and *Il1β* gene expression in the retina of the C57BL6/J (WT) eyes after spHOTs. A rapid increase in *Il1β* was detected as early as 6 h post-spOHT, peaked at 12 h, and remained elevated at 24 h (Figure 1C). Relative to the baseline, the levels of *Il1β* in the vitreous increased by 3.8-, 16.8-, and 18.7-fold at 6, 12, and 24 h post-spOHT, respectively. The *Il1β* levels in eyes after the stOHT were not significantly elevated, showing changes of 1.15-, 0.83-, and 1.0-fold relative to the corresponding baselines at 6, 12, and 24 h post-stOHT (Figure 1C).

To confirm the absence of ischemic conditions in the spOHT-challenged retinas, we measured the expression of the gene for ischemia marker protein, HIF1α, which was not elevated after either stOHT or spOHT. However, it was significantly upregulated in retinas that underwent IR (110 mm Hg IOP for 45 min) (Figure 1D), indicating that spOHT is a non-ischemic event.

### 3.2. SpOHT Induce Functional and Structural Damage to RGCs

OHT-induced inflammasome activation has been previously shown to inversely correlate with the electrophysiological responsiveness of RGCs [54]. We measured PERG to assess RGC function after spOHT and stOHT (Figure 2A,B). SpOHT induced a persistent, but not progressive, decrease in the mean PERG amplitude, (35.57 ± 8.13%, 31.07 ± 7.83%, and 37.28 ± 9.83%) as compared to the baseline at 7, 14, and 21 days after the spOHT, respectively (Figure 2C). A progressive increase in PERG latency was also observed, averaged at 6.43+/−4.34%, 8.37+/−2.92%, and 16.20+/−6.97% at 7, 14, and 21 days post-spOHT, respectively (Appendix A). These data suggest that the impairment of RCG electrophysiological function is long-lasting and not reversible. SpOHT treatment also resulted in a significant RGC loss, averaging at 20.41 ± 4.83% (*p* < 0.01) on day 7 post-spOHT, 20.4 ± 4.83% on day 14, and 10.52 ± 11.73% on day 21. RGC loss in stOHT-treated eyes, on the other hand, was minimal (Figure 2D). These data indicate that IOP elevation to 40 mm Hg has dramatically different impacts on RGC function and survival when applied as seven repetitive 1 min spikes vs. a single 7 min spike. 

### 3.3. Inflammasome Activity Coordinates Retinal Neuroinflammation

We used *Casp1*^−/−^ mice with a genetic inactivation of canonical inflammasomes, *Nlrp1b*^−/−^ mice with a targeted inactivation of the neuron-specific inflammasome [55,56], and *Gsdmd*^−/−^ mice with the ablation of the gasderminD pore-forming protein, the IL-1β release conduit, in many cell types, including CNS neurons and glia [34,57,58]. The ELISA data showed that *Casp1*^−/−^ and *GsdmD*^−/−^ mice had a complete blockade of intravitreal IL-1β release levels (a non-significant 1.35-fold and 1.6-fold increase vs. 12.1-fold increase in WT, relative to corresponding pre-injury baselines, respectively) at 24 h post-spOHT. Importantly, the spOHT-induced levels in these knockout strains were not significantly different from those in the WT sham-treated eyes and the naïve WT control eyes challenged with stOHT. The *Nlrp1*^−/−^ eyes, however, did not show any suppression of IL-1β release (14.4-fold increase) relative to its baseline, but the mean IL-1β release level in the vitreous of *Nlrp1*^−/−^ eyes was 2.7-fold lower than that in WT eyes after spOHT (Figure 3). 

The measurements of the concurrent release of IL-1β, TNFα, and CCL2/MCP-1 cytokines in the same samples performed as SimplePlex assays showed strong co-regulation between the release of CCL2/MCP-1 and IL-1β in *Casp1*^−/−^ and *Gsdmd* ^−/−^ mice, but not in *Gsdmd* ^−/−^ or *Nlrp1*^−/−^. Thus, the release of TNFα was significantly suppressed in *Casp1*^−/−^, *Gsdmd*^−/−^, and *Panx1*^−/−^, while *Nlrp1*^−/−^ showed only partial suppression. These data indicate that the global increase in IL-1β is largely controlled by the activities of the canonical Casp1 inflammasomes and GsdmD cytokine release pores.

### 3.4. Panx1 Is the Mechanosensitive Regulator of the Inflammasome

Panx1 is a cell surface mechanosensory Ca^2+^-permeable channel protein implicated in the pathophysiology of RGCs in retinal ischemia–reperfusion injury and glaucoma [33,34,39,54]. It is known to become activated via membrane stretching [59], like in the spOHT model used in this study. In the retina, it is expressed in many cell types, but is particularly enriched in RGCs [33]. To investigate the potential role of Panx1 in the spOHT-induced injury to RGCs, we used the *Panx1*^−/−^
*Casp11*^+/+^ mouse strain with a global genetic ablation of Panx1 and normal activity of CASP11 [60]. Consistent with the relative suppression of genes encoding for IL-1β, TNFα, and CCL2 cytokines in the retinas of *Panx1*^−/−^ mice, we found no significant increase in the release of IL-1β, TNFα, and CCL2 in the vitreous samples from the *Panx1*^−/−^ mice 24 hrs after sp-OHTs (Figure 3 and Figure 4), which contrasted with significant upregulation in response to the spOHT challenge in WT eyes. Among the inflammasome structural genes, the *Panx1*^−/−^ retinas showed significant suppression of the *Casp1* and *Nlrp3* genes (Figure 4); no significant suppression of the *gsdmd* gene was observed in these retinas. 

Using electrophysiological PERG recordings, we showed that the inactivation of either the *Casp1*, *Nlrp1*, or *Gsdmd* genes significantly suppressed the rate of OHT-induced decline in PERG amplitude relative to the pre-injury baselines in the WT spOHT-challenged mice (Figure 5A). The between-group comparison showed no statistically significant decrease in PERG amplitude at 7 dpi in the experimental eyes of *Casp1*^−/−^ (1.77 ± 9.69% N = 12 *p* = 0.99), *Nlrp1*^−/−^ (10.33 ± 8.43% N = 14; *p* = 0.89), *Gsdmd*^−/−^ (−24.38 ± 9.96% N = 11 *p* = 0.13), and *Panx1*^−/−^ (18.73 ± 3.28% N = 7 *p* = 0.55). In addition, the PERG latency at 7 dpi increased, although insignificantly, in *Casp1*^−/−^ (2.83 ± 3.78% *p* = 0.45), *Nlrp1*^−/−^ (4.49 ± 2.56% *p* = 0.1), *Gsdmd*^−/−^ mice (11.06 ± 5.72% N = 8 *p* = 0.13), and *Panx1*^−/−^ (3.70 ± 4.22% N = 8, *p* = 0.28) eyes following the spOHT challenge. These results demonstrate that RGC dysfunction after the spOHT challenge requires the activities of *Casp1*, *Nlrp1*, and the substrate *Gsdmd*, which is known to be highly induced by these neurons after acute OHT challenge [34].

### 3.5. Inactivation of the Inflammasome Protects RGCs 

To test the hypothesis that inflammasome activity plays a key role in RGC pathology after spiking IOP elevation, we first analyzed changes in the PERG amplitude and RGC density. We compared the changes in the *Casp1*^−/−^, *Nlrp1b*^−/−^, *Gsdmd*^−/−^, and *Panx1*^−/−^
*Casp11*^+/+^ mice with that in the WT at 7 days in the spOHT and stOHT injury models. The rate of the spOHT-induced changes in PERG amplitude was presented as relative to their pre-spOHT baselines (Figure 5A). The between-group comparison showed statistically non-significant changes in PERG amplitude: 1.77 ± 9.69% in the *Casp1*^−/−^ mice (N = 12 *p* = 0.99), 10.33 ± 8.43% in the *Nlrp1b*^−/−^ mice (N = 14, *p* = 0.89), 24.38 ±9.96% in the *Gsdmd*^−/−^ mice (N = 11, *p* = 0.13), and 18.73 ± 3.28% in the *Panx1*^−/−^ mice (N = 7, *p* = 0.55). In addition, changes in PERG latency were insignificant in the *Casp1*^−/−^ (2.83 ± 3.78%, *p* = 0.45), *Nlrp1b*^−/−^ (4.49 ± 2.56%, *p* = 0.1), *Gsdmd*^−/−^ mice (11.06 ± 5.72%, N = 8, *p* = 0.13), and *Panx1*^−/−^ mice (3.70 ± 4.22%, N = 8, *p* = 0.28). These results demonstrate that RGC dysfunction after the spOHT challenge requires the activities of NLRP1, CASP1, and its substrate GSDMD, which has been shown to be highly induced by these neurons in the acute OHT challenge [34]. 

SpOHT-induced RGC loss was studied in *Casp1*^−/−^, *Nlrp1b*^−/−^, and *Gsdmd*^−/−^ mice. RBPMS-positive RGCs were counted in retinas harvested 7 days after spOHT or stOHT challenges. The rate of RBPMS+ RGC loss in the experimental vs. naive age-matched retinas was significantly higher following spOHT (20.41 ± 4.83% at 7 dpi, 23.57 ± 4.8% at 14 dpi, and 10.52 ± 11.7% at 21 dpi, *p* < 0.01) (Figure 5A). No significant loss was observed after stOHT (6.95 ± 4.58%, *p* = 0.21 change in RGC density at 7 dpi, −0.05 ± 7.96% at 14 dpi, and 0.01 ± 11.23% at 21 dpi) (Figure 5B). We then assessed RGC loss in knockout mice with the inactivation of the inflammasome. The change in RGC density relative to that in naïve sham-operated eyes of the same genetic background at 7 dpi averaged at 7.71 ± 3.89% in the *Casp1*^−/−^ mice, 5.89 ± 4.1% in the *Nlrp1b*^−/−^ mice, and 4.65 ± 2.46% in animals with the inactivation of *Gsdmd*. Similar to the *Nlrp1b*^−/−^ retinas, the *Gsdmd*^−/−^ retinas were significantly protected from dysfunction and fully protected from structural loss (Figure 5A,B). In comparison, the rate of RGC loss in the WT mice following spOHT was 20.41 ± 4.83% (*p* < 0.01) (Figure 5B), whereas no significant RGC loss was observed in the eyes after stOHT (6.95 ± 4.58%, *p* = 0.21) (Figure 5B).

Protection via the inhibition of Casp1 or other inflammasome-signaling proteins could suggest that RGC death in this model occurred via pyroptosis, like in the retinal ischemia–reperfusion model [34]. However, inflammasomes can facilitate casp3-mediated apoptotic death via the combination of the CASP3-mediated disruption of GSDMD pore formation, resulting in apoptotic death, or the gasdermin-mediated release of mitochondrial cytochrome C that activates CASP3 [61,62]. We therefore tested whether CASP3 activity is essential for spOHT-induced cell death using the caspase-3-specific inhibitor, DEVD. When intravitreally injected 1 hr prior to spOHT in WT mice, DEVD completely protected RGCs (Figure 5B).

### 3.6. Inner Retina Cell Types That Activate Inflammasome

Two complementary approaches were used to determine the retinal cell types that activate inflammasomes in the spOHT injury via the detection of (1) proteolytically activated caspase-1, using intravitreally injected FAM-FLICA® Caspase-1 660substrate at 1 h prior to end points, or (2) the in vivo formation of citrine-labelled ASC speck, using the bioindicator ASC-citrine mice expressing the ASC core inflammasome scaffold protein fused to the green fluorescent marker citrine [34,41]. The analysis of retinal wholemounts for cell types that became fluorescently labelled due to activity of FAM-FLICA® Caspase-1 660substrate showed CASP1^+^ cells in the ganglion cell layer (GCL) of spOHT-challenged retinas, which were identified as RGCs with RBPMS^+^ co-staining (Figure 6C). Next, we used the same approach to detect the activities of pro-apoptotic caspase-8 and caspases-3/7 using the intravitreal co-delivery of FAM-FLICA Caspase-8-550 and FAM-FLICA Caspase-3/7 Alexa 488 substrates. In the GCL, active CASP8-labelled cells co-localized with CASP1^+^ RBPMS^+^ RGCs (Appendix A). CASP3/7-specific activity among the inner retinal cell types was detected in (1) RBPMS^+^ RGCs, (2) GFAP^+^ astrocytes, and (3) blood vessel endothelial cells (elongated nuclei aligned with blood vessels). However, neither astrocytes nor blood endothelial cells were positive for CASP1 activity in the spOHT-challenged retinas at 6 h post-injury (Figure 6A). These data show that the co-activation of pro-inflammatory CASP1 with pro-apoptotic CASP 3/7 and CASP8 occurs exclusively in RGCs at the early post-injury stage. In the control stOHT-challenged retinas, CASP1+ and CASP8+ RGCs were not detected (Figure 6B). Some CASP3/7+-positive cells were detected among GFAP^+^ astrocytes and, likely, blood vessel endothelial cells (characteristic elongated nuclei aligned along blood capillaries), but CASP1^+^ cells were absent in these samples (Figure 6B). 

Next, we used a co-localization analysis of retinal multiplex immunostaining to determine cell types that activate the major pro-inflammatory cytokines and the major retinal inflammasome complexes, NLRP1, NLRP3, and AIM2, that we previously characterized in the inner retina [34]. In the spOHT samples, specific labeling for IL-1β co-localized to neurons in the INL and was particularly strong in the GCL layers of the inner retina (Figure 7). The TNFα-specific labeling was confined to distinct cell types, including GFAP^+^ astroglia (arrows), CD11b^+^/Iba1^+^ microglia, and infiltrating CD45^+^/CD11b^+^ monocytes/macrophages (white arrowheads, Figure 7) in the retina 48 h after the spOHT challenge. Importantly, the citrine-positive ASC specks, representing the most active inflammasomes in the retina, co-localized with the GFAP^+^/TNFα^+^ astrocytes at the inner surface of the retina (yellow arrows, Figure 7). In contrast, retinas from the sham and steady OHT controls had IL-1β- and TNFα-specific labeling that was significantly reduced, along with a lack of infiltrating monocytes (Figure 7 bottom panel). These data indicate that, in contrast to a steady non-ischemic OHT stress, the repetitive spOHT robustly induce acute inflammasome activation that is restricted to inner retinal neurons, particularly RGCs in the inner retina. 

Previous research identified three canonical inflammasomes in the retina: NLRP1, NLRP3, and AIM2, with few reports showing the activation of NLRC4- and NLRP12-containing complexes [24,34,35,63]. We analyzed the cellular localization of the canonical complexes via immunostaining in thin slices of retinas 48 h after spOHT and stOHT challenges. These data showed an increase in NLRP1 immunoreactivity in the GCL, co-localizing with TUJ1^+^ RGCs and their axons after spOHT, which was significantly stronger than in the sham-operated controls (Figure 8). The NLRP3-specific labeling co-localized with some RGCs and glial cells in the sham controls, but showed little change after the spOHT challenge (Figure 8). Both NLRP1 and NLRP3 sensors were also expressed by infiltrating monocytes (yellow arrowheads, Figure 8). The AIM2 inflammasome-specific labeling co-localized exclusively with glutamine synthetase-positive (GlutSyn^+^) Muller glia and also showed an increased immunoreactivity in spOHT-challenged retinas, relative to the stOHT controls. The infiltration of monocytes/macrophages co-expressing CD45^+^ and CD11b^+^ markers was detected in the retina and optic nerve of the spOHT-challenged eyes, but not in the stOHT- or sham-treated controls (Figure 9). These cells were numerous at the inner retina surface, in the GCL and IPL layers, and the optic nerve after repetitive spOHT-induced injury (marked by yellow arrowheads in Figure 9), indicating a contribution of the adaptive immune system to the pathogenesis.

### 3.7. Inflammasome Activation in Glaucoma

To answer the question of whether inflammasomes play a pivotal role in chronic OHT/glaucoma, as in the spOHT stress, we utilized the Ad5-MYOC-induced chronic OHT glaucoma model, developed by Grotegut and Kuehn [46,47] and characterized by a reproducible pattern of early-onset mild IOP elevation. We correlated IOP dynamics in this model with changes in PERG amplitude and IL-1β activity in the vitreous body at 3, 5, and 8 weeks after OHT induction. As expected, the PERG amplitude declined significantly, starting at 2 weeks post Ad5-MYOC injection in the WT eyes (Figure 10A) in strong correlation with the IOP increase (Figure 10B). In the WT eyes, we observed a significant increase in the levels of vitreous IL-1β at these time points, peaking at 5 weeks at 2.5 ± 0.5 pg/mL (Figure 10C). When compared to the maximum IL-1β elevation in the positive control eyes, injected with 3 mM ATP- or challenged with 1 h ischemia–reperfusion, vitreous IL-1β levels in the Ad5-MYOC-injected eyes were elevated approximately by 38% and 44% at 3 weeks, 67% and 59% at 5 weeks, and 18 and 16% at 8 weeks post-injection, respectively. This persistent cytokine “tide” in this glaucoma model correlated with a 29.8 ± 3.5% loss of RGCs at 8 weeks post-OHT induction (Figure 10D). The analysis of the ASC speck formation to detect retinal cell types that activate inflammasomes in the chronic OHT-induced glaucoma showed that the RBPMS^+^ RGCs accounted for the majority of ASC-citrine-positive cells (≥60%) in retinal wholemounts, whereas the second most common type were GFAP^+^ astrocytes (~25%), and only a minor (<10%) proportion of ASC specks co-localized with retinal blood vessels (Figure 10E–G). Combined, these data indicated that RGCs and astrocytes are the two key types of inner retinal cells contributing to IL-1β release into the vitreous fluid during the progression of OHT-induced glaucoma. 

## 4. Discussion

In this study, we investigated the impact of immune responses induced by repetitive (spiking) non-ischemic IOP elevations on the function and viability of RGCs in mouse eyes. We demonstrated that, in contrast to stOHT of a similar magnitude and duration, the spOHT challenge caused the rapid activation of the inflammasome in the retina, resulting in the release of pro-inflammatory cytokines. These events preceded an irreversible dysfunction and caspase-3-dependent loss of RGCs. A lower-grade activation of the inflammasome was also observed 3 weeks post-induction in retinas challenged with chronic OHT in the Ad5-MYOC-induced glaucoma model. Despite major differences in the types of OHT insult in these models, both spOHT and chronic OHT challenges induced RGC dysfunction and death, whereas its inactivation prevented RGC deficits in both models. Our results suggest the mechanistic involvement of CASP1, CASP8, NLRP1, NLRP3, GSDMD proteins, and inflammatory cytokines expressed by neuronal, glial, and infiltrating immune cells in the spOHT-induced RGC pathology. 

### 4.1. Acute and Progressive Inflammasome Activation in the spOHT Model

In the spOHT model, the production of pro-inflammatory cytokines, TNFα, IL-1β and monocyte chemoattractant protein (MCP-1 or CCL2) was dependent on the activity of the Panx1-inflammasome signaling axis. We obtained evidence of the mechanistic involvement of the endogenous NLRP1 inflammasome, as well as its downstream targets GSDMD and CASP1 in the spOHT-induced pathophysiology of RGCs, by using gene knockout strains and drug inhibitors. In addition, we also confirmed the essential role of the upstream mechanosensitive regulator of NLRP1 and NLRP3 and the PANX1 channel. RGC dysfunction and loss strongly correlated with activation of NLRP1, NLRP3, and their products, CASP1, IL-1β cytokine, and GSDMD. Importantly, the ablation of the *Nlrp1b* gene was profoundly neuroprotective in the chronic model (Figure 10D). The alternative IL-1β convertase, CASP8, which is both pro-inflammatory and pro-apoptotic via the activation of CASP3, was also synergistically co-activated with CASP 1 and in RGC cells (Appendix A). Finally, AIM2, known to drive pyroptotic death and sensitized by intracellular self-DNA [64,65], was specifically active in Muller glia, the cell type that was also implicated in the glaucomatous loss of RGCs [66,67]. Significantly, in the control stOHT-challenged eyes, we did not detect any significant increase in inflammasome activity (release of cytokines), the induction of CASP 1, CASP 8, and damage of RGC, thus allowing us to conclude that the spiking pattern of OHT is a key trigger of retinal inflammation and injury. To summarize, our model provided new mechanistic insights into (1) very early post-spOHT pro-inflammatory events, and (2) mechanisms driving excessive neurotoxicity after repetitive vs. single, non-repetitive IOP spikes. Finally, mechanistic similarities between acute spOHT and chronic OHT-induced glaucoma, including the inflammasome/CASP1-dependent RGC dysfunction and death, suggest that spOHT can be instrumental in studies of the earliest pathophysiological events in the OHT-challenged retina.

### 4.2. Mechanosensor Signaling Drives Inflammasome Activation

Most inflammasomes are induced by a combination of internal (loss of ionic and energetic homeostasis) and external signals via surface receptors. These are integrated into Signal 1, which facilitates the transcriptional induction of inflammasome components via TLR, TNF, IL-1, C5a receptors, and alarmins, and Signal 2, which facilitates complex assembly in response to lysosomal damage and extracellular ATP and K^+^ elevations via Panx1-P2X4/7 activation [68,69,70,71,72]. Abundant evidence from the literature supports a model in which the spOHT-induced stimulation of the mechanosensitive Ca^2+^ channels, Panx1 [33,73,74], Piezo-1 [75], and transient receptor potential (TRP) vanilloid TRPV1 and TRPV4 [3,76,77] channels in neural cell plasma membrane initiates pro-inflammatory signaling [66,78,79,80]. Although only Piezo1 was confirmed as an interaction partner of Panx1 [39,81,82], it is feasible to suggest that a synergistic activation of all these channels through repetitive IOP spikes exacerbates Panx1 opening due to the increase in intracellular Ca^2+^, causing the over-activation of the inflammasome due to subsequent ATP release [34]. In support of the key role of mechanosensory Panx1 signaling in the retina, recent reports showed its pivotal role in both acute ischemia-induced and glaucomatous degeneration, where its activity strongly correlated with the induction of the inflammasome and production of IL-1β [33,54]. Consistently, our results showed that the suppression of inflammation and the protection of RGC via a *Panx1* blockade was similar in efficiency to the ablation of the *Casp1* and *Gsdmd* genes.

### 4.3. The Neurotoxic Pathways Downstream of the Inflammasome 

The neurotoxic pathways downstream of the inflammasome include pyroptotic and non-pyroptotic extracellular paracrine pathways, such as the GSDMD-NT pore-mediated release of IL-1β, several types of alarmins, miRNAs, and ASC speck complexes [83,84,85], as well as the cleavage–activation of fluorescent FAM-FLICA CASP1 and CASP8 substrates. Most of these pathways facilitate strong pro-inflammatory responses via local glial activation, blood neutrophil attraction, tissue migration [86,87,88,89,90,91], and spreading inflammation [92]. In particular, the activation of GsdmD-NT pores has previously been shown to mediate cytokine release that sustains and propagates proinflammatory signaling in the retina in a way that is similar to a “cytokine storm” during sepsis or viral infection [93,94].

Furthermore, similar to other gasdermins, GsdmD activation by the inflammasome was reported to facilitate mitochondrial damage [95,96] and pro-apoptotic signaling via the release of mitochondrial DNA and cytochrome C [61,95,97]. Since mitochondrial dysfunction and the subsequent loss of RGC functionality and viability are the key pathological events in cerebral ischemia [98] and glaucoma [99,100,101], we obtained direct evidence of GsdmD involvement in the dysfunction and death of RGCs using GsdmD^−/−^ retinas. Overall, because functional and structural damage to RGCs was prevented by the ablation of the *Nlrp1*, *Panx1*, or *Gsdmd* genes, we concluded that the NLRP1 and NLRP3 inflammasomes, expressed by RGCs and macrophages, are key mediators of neurotoxicity in the spOHT injury model. 

Finally, the high level of neuroinflammation and cytokine release strongly correlated with the infiltration of blood monocytes. The concurrent release of IL-1β, TNFα, and particularly MCP-1/CCL2 cytokines, observed in the vitreous and retina of the spOHT-challenged eyes, is also known to weaken the blood–retina barrier and promote the infiltration of blood-borne monocytes and leukocytes elsewhere [45,102,103]. Such infiltration was evident at 48 h post-OHT induction, as confirmed by abundant CD45^+^ CD11b^+^ cells in the retina and optic nerve (Figure 9), and strongly correlated with high levels of cytokines that were not initially expected in such a mild injury. As a matter of fact, monocyte and leukocyte infiltration are implicated in various retinal degenerations, including OHT-induced ischemia–reperfusion [104], AMD [105], and glaucoma [86,106]. The contribution of both endogenous neural cells and infiltrating monocytes to the over-activation of inflammasomes and the release of cytokines after spOHT challenge call for a deeper dissection of the mechanisms underlying OHT-induced injuries at the cell-type level. Innovative single-cell RNA sequencing and spatially resolved transcriptomics technologies will provide such a resolution in our future experiments. 

### 4.4. Pyroptosis vs. Apoptosis

The activation of the inflammasome end products CASP1 and GSDMD in response to the spOHT challenge suggested cell death via pyroptosis [107,108]; however, we used FAM-FLICA Casp1/3/8 labeling and a DEVD blockade of CASP3 activation to obtain evidence that the spOHT challenge triggered RGC loss via apoptosis. This conclusion was made despite our data indicating the key roles of CASP1 and GSDMD in RGC death, which are typically associated with pyroptotic death, as we reported earlier in the retinal ischemia model [34]. The molecular mechanism for the CASP1- and CASP8-dependent activation of CASP3 is well characterized [61,62,109]. Moreover, the most recent insights into the non-canonical functions of GSDMD and CASP8 showed a non-pyroptotic GSDMD–ASC–CASP8 interaction as well as the alternative cleavage of GSDMD by activated CASP3 into a non-pore-forming fragment [59]. Both of these events prevent pyroptosis and facilitate apoptosis in a heightened inflammatory environment. In support of this mechanism, we observed the co-activation of CASP1 and CASP3/7, specifically in RGCs in the spOHT-challenged retinas (Figure 6), which was blocked in either *Casp1*^−/−^ retinas or in WT retinas treated with the Casp1 inhibitor VX-765, both of which protected RGCs functionally and structurally (Figure 5). It is reasonable to conclude that the neurotoxic activity of Casp1 is directly essential for endogenous OHT-induced RGC death, since it is present in RGCs directly following injury, and its ablation caused a significant inhibition of IL-1β release into the vitreous, resulting in RGC protection [34,54]. At the same time, the activity of retinal CASP1 may also affect the viability of RGCs indirectly, since the bulk of CASP1-containing inflammasomes are not RGC-specific and are activated by astrocytes, microglia, and retina-infiltrating immune cells. 

## 5. Conclusions

In conclusion, repetitive IOP spikes in the spOHT model induce an acute dysfunction and degeneration of about 1/5 of the RGC population via the activity of the canonical inflammasomes and GSDMD. The extent of degeneration and RGC death were similar to that observed in most models of OHT-induced glaucoma [110,111,112,113]. Our results provide evidence of the mechanistic involvement of inflammasomes from both endogenous and blood-borne cells in the pathophysiology of RGC loss after repetitive spiking IOP. Our results suggest that repetitive IOP spikes can both initiate de novo or accelerate an ongoing retinal pathology. As suggested in this and other studies [45,103], post-spOHT inflammatory damage is unexpectedly high and can serve as a facilitating risk factor in predisposed individuals, including those with lowered arterial blood pressure [114], systemic diseases, advanced age, pre-existing glaucoma, and NTG patients [115,116,117], a certain proportion of which suffer from IOP spike-induced damage [118]. Current attempts to control IOP spikes in clinics using conventional IOP-lowering drugs have failed [119], indicating a need for neuroprotective strategies targeting inflammasome-mediated mechanisms of injury.

## Figures and Tables

**Figure 1 cells-12-02626-f001:**
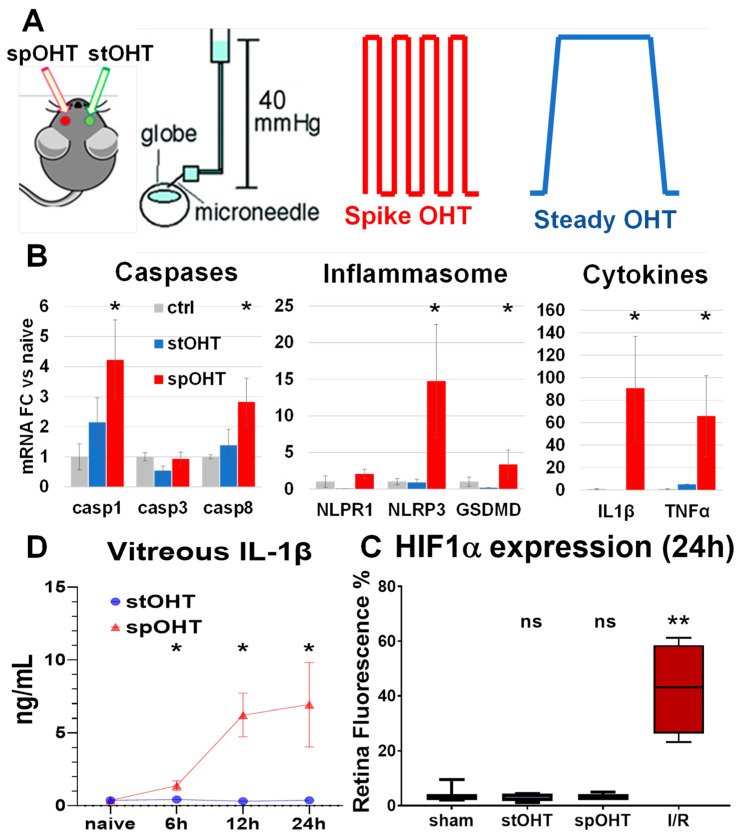
Repetitive IOP spikes induce the activation of inflammatory biomarkers and RGC dysfunction and loss. (**A**) Experimental paradigm of the spiking OHT (spOHS) model and steady OHT (stOHT) control; (**B**) qRT-PCR data on relative changes in transcript abundances of inflammasome pathway genes in retinas. Mean fold change vs. naïve ± SE, * *p*< 0.05, *n* = 3–5. (**C**) Intravitreal IL1-β measured at 6, 12, and 24 h after spOHT (red) and stOHT eyes (OD, blue). Mean ± SE, * *p* < 0.05, *n* = 5. Negative control: *Casp1*^−/−^ eyes (Cs1KO). (**D**) Quantification for HIF1a fluorescent labeling in retina wholemounts from eyes exposed to sham, stOHT, spOHT, or IR. Mean percentile change vs. sham ± SE, ** *p* < 0.05, *n* = 5.

**Figure 2 cells-12-02626-f002:**
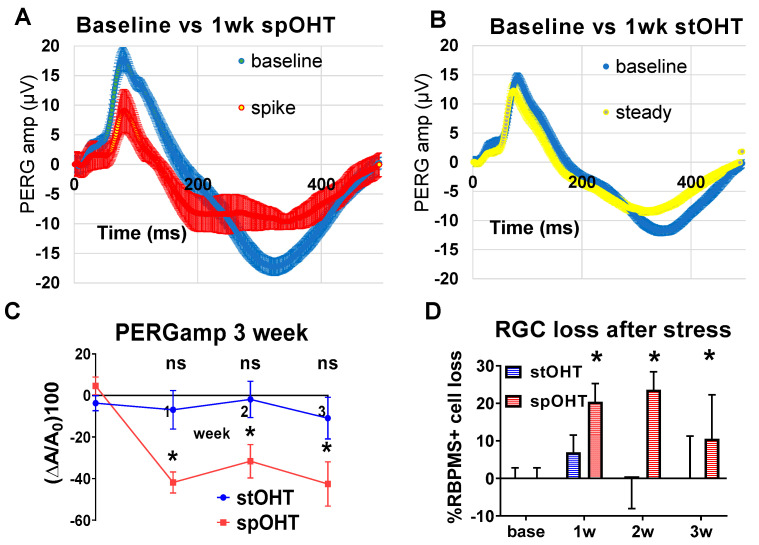
PERG changes after a single spOHT challenge are stable over time. Electro-physiological recordings of changes in PERG amplitude in the retinas exposed to either spOHT stress or stOHT (green bars) elevation for 7, 14, and 21 dpi. * *p* < 0.05. (**A**) Representative PERG grand average waveforms recorded in WT eyes prior to (baseline, blue) or 7d (red) after the challenge. (**B**) Representative PERG grand average waveforms recorded in WT eyes at 7 d after either spOHT (red) or stOHT (yellow) challenges. (**C**) Dynamics of PERG amplitude at 7, 14, and 21 dpi in C57Bl6/J eyes after spiking vs. steady OHT challenges. Mean ± SE, * *p* < 0.05. (**D**) SpOHT-induced RGC loss at 1–3 weeks after spike or steady OHT. Mean ± SE, * *p* < 0.05.

**Figure 3 cells-12-02626-f003:**
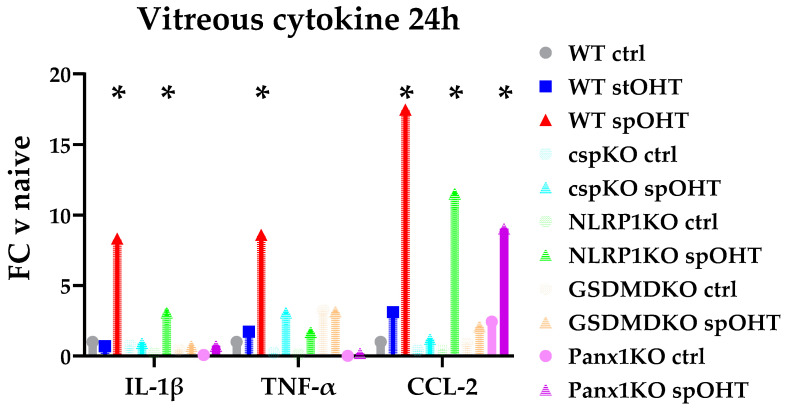
Cytokine release into the vitreous strongly correlate with inflammasome activity. Cytokine levels in the vitreous of WT, *Nlrp1*^−/−^, *Casp1*^−/−^, *Gsdmd*^−/−^, and *Panx1*^−/−^ mice were measured simultaneously via multiplex ELISAs at 24 h after spOHT. The mean intravitreal concentration was measured as pg/mL (±SE) in a given transgenic mouse line and normalized using internal standards for every plate to allow between-plate comparison.* *p* < 0.05, *n* = 3–5.

**Figure 4 cells-12-02626-f004:**
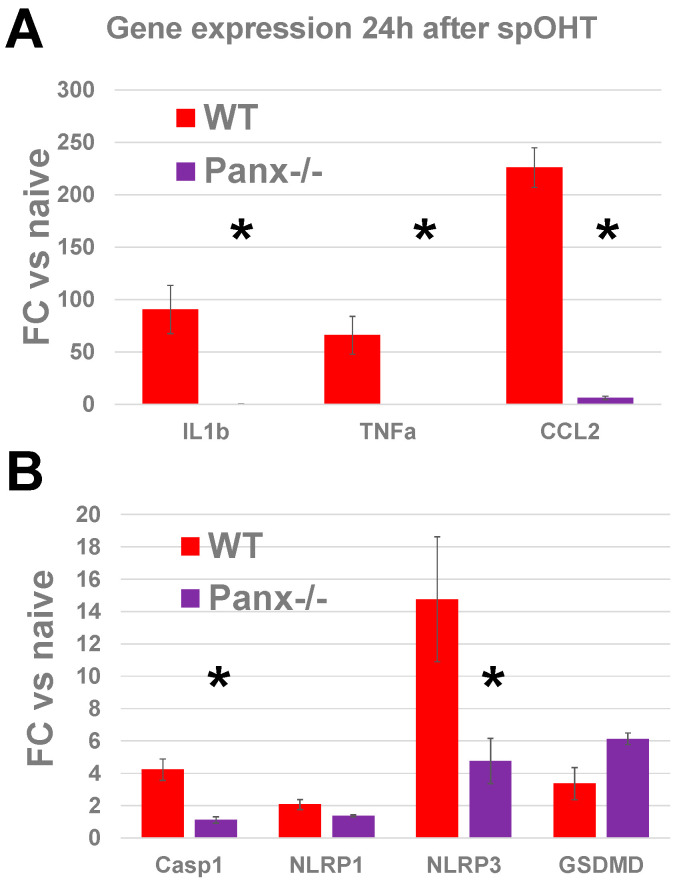
Panx1 is essential for the transcriptional activation of cytokines and inflammasome pathways in response to spOHT. Gene expression analysis was performed via RT-PCR in retinal samples collected 24 h after spOHT. Significant increases in *Il1b*, *Tnf*α, *Ccl2*, *Casp1*, *Nlrp3*, and *Gsdmd* transcripts found in WT mice ((**A**), red) were blocked by PANX1 channel inactivation in *Panx1*^−/−^ retinas (**A**) purple). *Panx1* inactivation did not affect the upregulation of the *gsdmd* gene (**B**). * *p* < 0.05, *n* = 3–5.

**Figure 5 cells-12-02626-f005:**
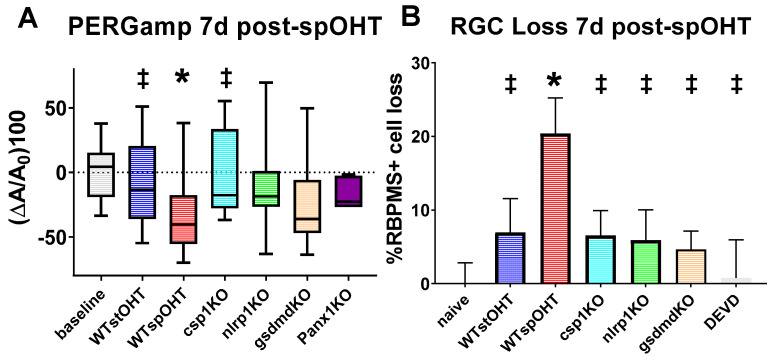
PERG and RGC after spOHT injury with inflammasome knockout. (**A**) significant PERGamp deficit was detected in WT and GsdmD^−/−^ eyes at 1 w after spOHT. No significant change in PERGamp was detected in spOHT-challenged eyes of the inflammasome-deficient strains *Nlrp1*^−/−^, *Casp1*^−/−^, and *Panx1*^−/−^ in WT eyes pre-treated with CASP3 inhibitor DEVD and in control WT eyes with stOHT challenge. (**B**) RGC loss is significantly suppressed via the inactivation of all key inflammasome proteins, the IL-1 release pore protein *Gsdmd*, or caspase 3. The density of RBPMS + RGCs was assayed through direct counting in wholemounts 7 d after OHT challenge. N = 10–15, * *p* < 0.05 vs baseline, ǂ *p* < 0.05 vs WTspOHT.

**Figure 6 cells-12-02626-f006:**
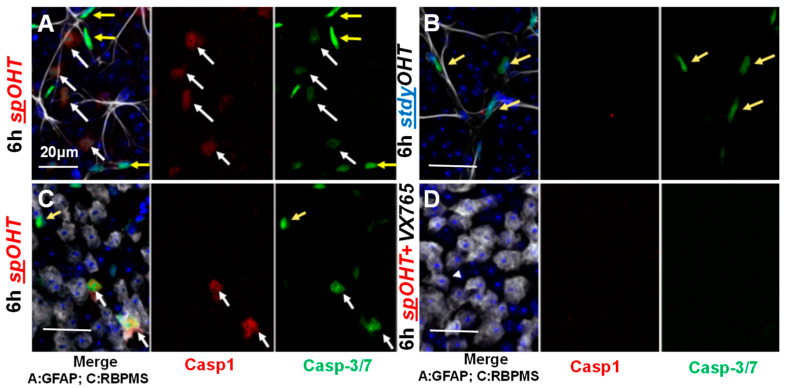
Acute caspase activation in the GCL of spOHT-challenged retinas. (**A**) CASP1 (red) and CASP3/7 (green) activities in wholemount retinas were detected via an in vivo injection of FAM-FLICA substrates. (**B**) FAM-FLICA labeling in control eyes after stOHT. Retinas were co-stained with GFAP (light grey, **A**,**B**) and RBPMS (light grey, **C**) white/merge to identify astrocytes and RGCs, respectively. White arrows indicate cells that co-activate CASP1 and CASP3/7; yellow arrows are pointing at cells with CASP 3/7 activity only. (**D**) Pre-treatment with VX765 casp1 blocked both CASP1 and CASP3/7 activity after spOHT. Size bar, 20 µm.

**Figure 7 cells-12-02626-f007:**
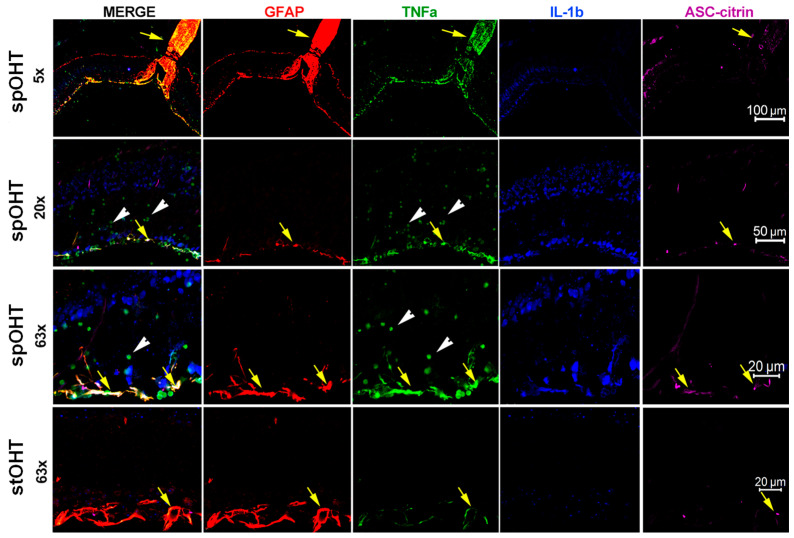
Pro-inflammatory cytokines in the spOHT-challenged retina. Immunolabeling for IL-1β (blue) co-localized with neurons in the GCL and INL; TNFα-specific labelling (green) confined to GFAP^+^ astroglia (arrows), infiltrating monocytes (arrowheads) in the retina 48 h after the spOHT challenge. ASC-citrine specks (magenta) co-localized with the GFAP^+^ TNFα^+^ astrocytes and Muller glia (yellow arrows) in the inner retina, and with astrocytes in the optic nerve. In control stOHT-treated retinas (bottom panels), both TNFα- and IL1β-specific labeling were reduced; no infiltrating monocytes were detected. Size bars in µm.

**Figure 8 cells-12-02626-f008:**
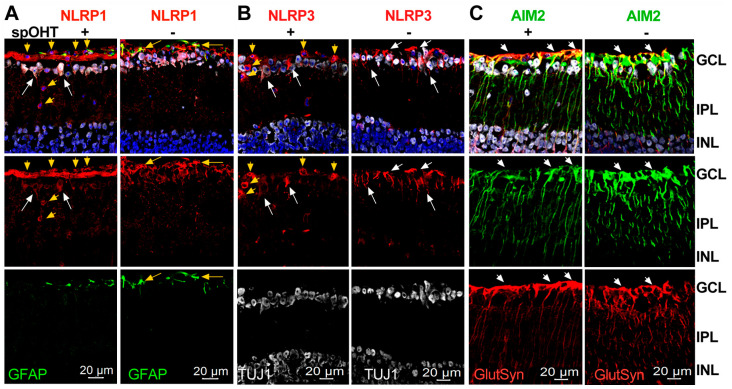
Inflammasome sensor proteins in the spOHT-challenged inner retina. (**A**) Immunolabeling specific for NLRP1 co-localized with TUJ1^+^ neurons, their axons, and GFAP^+^ astrocytes (yellow arrowheads) in the GCL of both spOHT- and sham-treated control retinas; in spOHT samples, NLRP1 was also detected in infiltrating monocytes (yellow arrowheads). (**B**) NLRP3 labeling co-localized with TUJ1^+^ neurons and infiltrating monocytes (yellow arrowheads) in the GCL of spOHT-challenged retinas; in sham controls, it also labeled glial cells (white arrows). (**C**) AIM2 labeling co-localized with GlutSyn+ Muller glia (white arrows) in both sham controls and spOHT retinas, and showed an increased accumulation in the GCL region after spOHT challenge. Size bars, 20 µm.

**Figure 9 cells-12-02626-f009:**
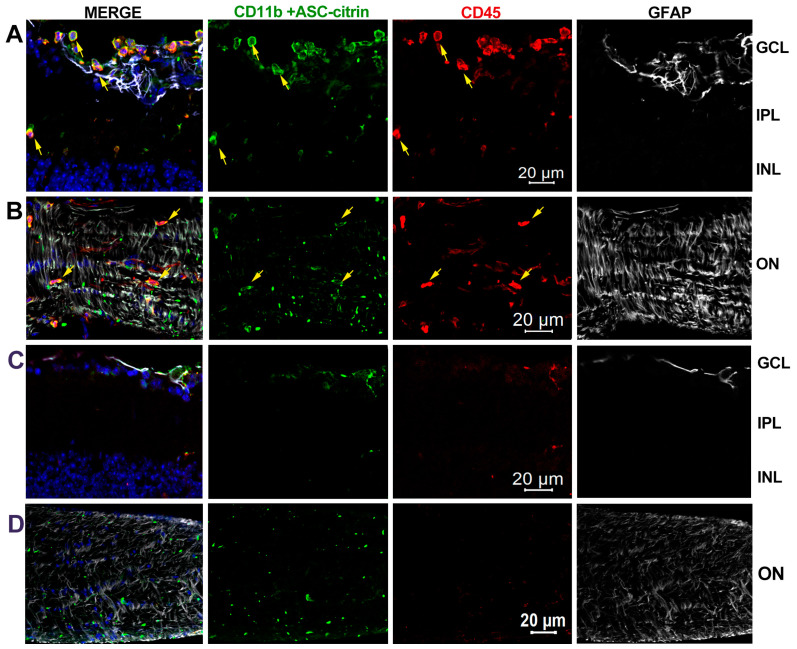
Monocyte/macrophage infiltration into the inner retina and optic nerves after spOHT challenge. (**A**,**B**) Representative micrographs for cells expressing CD45 (red) and CD11b (green) markers in rounded monocyte/macrophage cells that were abundant at the inner retinal surface, and in the GCL, INL, and optic nerves of the spOHT-challenged eyes. Co-staining with GFAP (light grey, top panels) identified astrocytes. (**C**,**D**) Control immunostaining in stOHT-challenged optic nerves showed no CD45^+^ monocytes in both retina and optic nerves. Small punctate CD45+ CD11b+ cells are ramified microglia. Bright green puncta represent citrine-labeled ASC specks of mature inflammasomes that are abundant in the optic nerve astrocytes and infiltrating cells (yellow arrows).

**Figure 10 cells-12-02626-f010:**
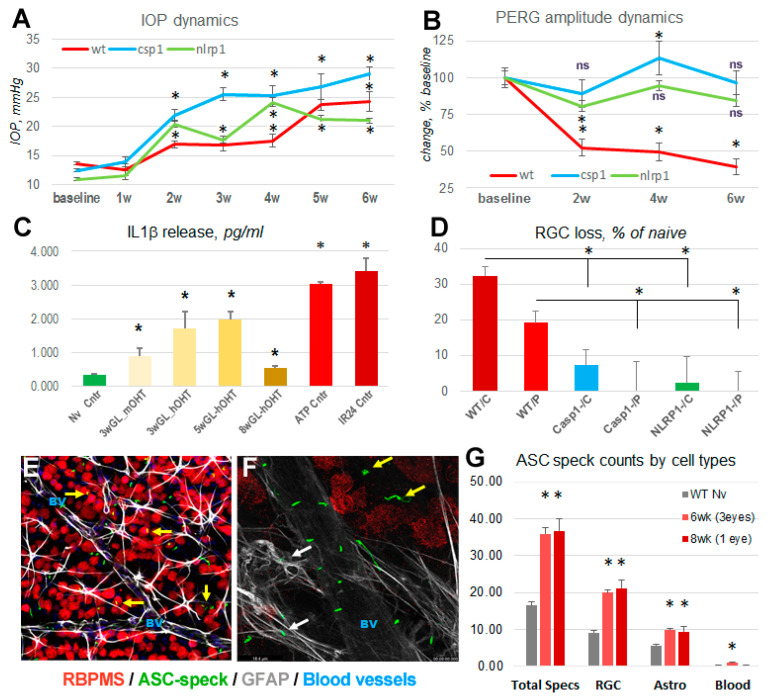
Inflammasome activation in the Ad5-MYOC-indiced chronic OHT/glaucoma model. Dynamics of initial IOP changes (**A**) was similar in WT, Casp1-, and NLRP1- strains, with relatively higher levels of IOP elevation in knockouts at 2–4 weeks. An increase in IOP in experimental eyes temporally correlated with RGC dysfunction, detected with a ~50% decline in PERGamp (**B**) in WT but not in CASP1- and NLRP1- mice at 2 weeks post-induction. (**C**) IL-1β release was detectable 3-8 weeks post-OHT induction, peaking at 5 weeks at ~60% of that in positive control eyes with injection of ATP or ischemia–reperfusion (IR) injury. (**D**) RGC density changes in experimental eyes vs. eyes from naïve controls showed 32% and 19.1% loss in WT center (/C) and periphery (/P). The loss was significantly less in CASP1- and NLRP1- retinas at 7dpi after spOHT challenge, * *p* < 0.05 (**E**,**F**). ASC speck labeling (green specks/filaments) revealed mature inflammasome complexes co-localizing with RBPMS+ (red) RGCs (yellow arrows), GFAP+ (grey) astrocytes (white arrows), and blood vessels (unstained dark grey structures, blue “BV”). (**G**) Quantification of ASC-citrine + cell types in retinal wholemounts at 6 and 8 weeks post-induction indicated predominant localization to RGCs and astrocytes, * *p* < 0.05.

## Data Availability

All data are contained within this article or Appendix A. Histology data used for cell counting are available upon request.

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
