# Peer review of "The Inflammasome-Dependent Dysfunction and Death of Retinal Ganglion Cells after Repetitive Intraocular Pressure Spikes"

_cells, 2023, doi:10.3390/cells12222626_

Round 1

Reviewer 1 Report

Comments and Suggestions for Authors

The authors investigated the upregulation of inflammasome-related genes such as NLRP1, NLRP3, AIM2, caspases -1, -3/7, -8, gasdermin D (GsdmD), and the release of interleukin-1β (IL-1β) and other cytokines in a transient non-ischemic OHT spikes (spOHT) and in an induced glaucoma model carrying a human Myocin gene mutation. The data are of high quality with multiple techniques employed. All my comments are minor and listed as follows.

1.    The authors may mention the gender (male, female or both) of the animals they used.

2.    For the real time PCR studies, the authors may add more information such as how much RNA or cDNA were used, the predicted PCR products length, the sequences for amplifying beta-actin or GAPDH, etc to conform to the MIQE guidelines.

Bustin SA, Benes V, Garson JA, Hellemans J, Huggett J, Kubista M, Mueller R, Nolan T, Pfaffl MW, Shipley GL, Vandesompele J, Wittwer CT. The MIQE guidelines: minimum information for publication of quantitative real-time PCR experiments. Clin Chem. 2009 Apr;55(4):611-22. doi: 10.1373/clinchem.2008.112797. Epub 2009 Feb 26. PMID: 19246619.

3.    Do the authors use single scan or multiple scans with Z-stack at ??? um for the confocal studies.

4.    The authors may cite some references for the specificity of the antibodies or the ELISA kits they utilized, because the omission of the primary antibodies or purchasing from commercial sources doesn’t guarantee the specificity of the antibodies.  

5.    The authors used some single KO mice, I was wondering if using double or triple KO mice, the neuroprotective effect may be more significant.

6.    The authors may give a tiny discussion regarding the usage of single cell NGS or proteomic, etc techniques to study the activation of inflammasome in their unique animal models.

7.    The authors studied a gap junction protein called pannexin1 in their studies, it is well established the neuronal gap junction, namely connexin36 is also highly expressed in retinal neurons and related to glaucoma.   

Thank you!

Author Response

Reviewer 1

  1. The authors may mention the gender (male, female or both) of the animals they used.

RESPONSE: Both male and female mice were used in this research. Equal numbers of each were assigned per group. This has been added to the Materials and Methods section of the manuscript.

  1. For the real-time PCR studies, the authors may add more information such as how much RNA or cDNA was used, the predicted PCR product length, the sequences for amplifying beta-actin or GAPDH, etc to conform to the MIQE guidelines.

RESPONSE:

We used 500ng RNA to make cDNA in 20ul reaction. For qPCR, we used diluted cDNA 1:10, and used 5 ul diluted cDNA dilution. For each pair, the predicted product length was controlled by PCR. These specifications have been added to the Materials and Methods.

  1. Do the authors use single scans or multiple scans with Z-stack at ??? um for the confocal studies.

RESPONSE: Single scans with 20x lens were used for the cell counts and for cross-section images. This information has been added to the Materials and Methods.

  1. The authors may cite some references for the specificity of the antibodies or the ELISA kits they utilized, because the omission of the primary antibodies or purchasing from commercial sources doesn’t guarantee the specificity of the antibodies.

RESPONSE: All primary antibodies used have been listed in the Supplemental materials.

  1. The authors used some single KO mice, I was wondering if using double or triple KO mice, the neuroprotective effect may be more significant

RESPONSE: We agree that using double/triple-KO mice could show more neuroprotective effects, however, we are not aware of such strains being readily available at the major depositories One mouse strain used in this study was, indeed, an equivalent of a double knockout: the Casp1 KO strain used in this study contains a deletion, inactivating  Casp4(11),  another inflammasome-regulated caspase. We have ruled out active Casp4(11) contribution to RGC injury in our previous studies, where we tested Casp4(11) single KO strain, but it showed no protection ( PMCID: PMC5895610 ). Follow link https://static-content.springer.com/esm/art%3A10.1038%2Fs41598-018-23894-2/MediaObjects/41598_2018_23894_MOESM2_ESM.doc

  1. The authors may give a tiny discussion regarding the usage of single-cell NGS or proteomic, etc techniques to study the activation of inflammasome in their unique animal models.

RESPONSE: We have added the following piece to the discussion suggesting the scNGS or proteomic studies could provide molecular insight into the activation of inflammasome in this model. The following comment was added to the discussion: “The contribution of both endogenous neural cells and infiltrating monocytes to the over-activation of inflammasomes and release of cytokines after spOHT challenge, call for a deeper dissection of mechanisms underlying OHT-induced injuries at the cell type-level. Innovative single-cell RNA sequencing and spatially resolved transcriptomics and proteomics technologies will provide such resolution in our future experiments.”

  1. The authors studied a gap junction protein called pannexin1 in their studies, it is well established the neuronal gap junction, namely connexin36 is also highly expressed in retinal neurons and related to glaucoma.

Dr. C. Green compared protection observed by blocking connexin36 in his recent review ( PMID: 33578721  ). He pointed out that the mechanism of neurotoxicity via these two family of channel proteins are different. We are planning to address this interesting question in our future studies.

Reviewer 2 Report

Comments and Suggestions for Authors

This is a very interesting manuscript by the authors demonstrating the contribution of inflammasome in the death of retinal ganglion cells in a model of repetitive spikes of ocular hypertension, unraveling some mechanisms in glaucoma.

The authors used b-actin or gapdh genes as housekeeping genes. Why were the genes chosen? in which conditions were used? are the authors sure that these genes are stable in all tested conditions?

It would be interesting to compare the HIF1alfa expression in the stOHT model. Did the authors perform this experiment?

The authors mentioned that RGC loss was assessed by counting RBPMS+ cells in 16 fields in 4 retinal quadrants in 3 regions of the same eccentricities. The graphs only show the average number. Is there any difference between retinal regions or eccentricities? Also, in figure 2D RGC loss was counted for 1, 2 and 3 weeks and in the methods it is mentioned 7 days. Can you calrify this issue?

The title of Fig 4A is 48h after spOHT but in the legend is referred 24hs.

Indicate the p value when referring “partial suppression” – line 306

In figure 7, the labels of the images should be the ones used throught the paper.

The authors demonstrate infiltration of immune cells to the retina. it would be interesting to evaluate the permeability of blood-retinal barrier in the spOHT model and identify the molecular players involved.

Author Response

Reviewer 2

  1. The authors used b-actin or gapdh genes as housekeeping genes. Why were the genes chosen? in which conditions were used? are the authors sure that these genes are stable in all tested conditions?

RESPONSE: β-actin and GAPDH are “housekeeping” genes constitutively expressed in almost all tissues in high amounts. The expression of β-actin and GAPDH are widely used as loading controls. In our model and in our hands, the expression of β-actin and and GAPDH is consistently stable across experimental groups.

  1. It would be interesting to compare the HIF1alfa expression in the stOHT model. Did the authors perform this experiment?

RESPONSE: Thank you for pointing this out. We performed additional experiments to investigate the expression of HIF1α. Results have been added to Fig. 1 in the revised manuscript.

  1. 3. The authors mentioned that RGC loss was assessed by counting RBPMS+ cells in 16 fields in 4 retinal quadrants in 3 regions of the same eccentricities. The graphs only show the average number. Is there any difference between retinal regions or eccentricities? Also, in figure 2D RGC loss was counted for 1, 2 and 3 weeks and in the methods it is mentioned 7 days. Can you calrify this issue?

RESPONSE: The counting was performed as described to account for possible differences between retinal regions. However, no significant differences were observed in percent or RGC loss between mid-retina and peripheral retina in specific regions. We changed to 1, 2, and 3 weeks in the Materials and Methods.

  1. The title of Fig 4A is 48h after spOHT but in the legend is referred to 24hs.

RESPONSE: Thank you for pointing out the typo of 48h in the title of Fig. 4A. It has been corrected to 24h.

  1. Indicate the p value when referring “partial suppression” – line 306

RESPONSE: As reflected in the graph of the data, there was no significant suppression of nlrp1 gene expression in the panx1 KO. The description of “Partial suppression” was incorrect. This has been corrected in the text in the revised manuscript.

  1. In figure 7, the labels of the images should be the ones used throughout the paper.

RESPONSE: Thank you for pointing it out. This mistake has been corrected

  1. The authors demonstrate infiltration of immune cells to the retina. It would be interesting to evaluate the permeability of blood-retinal barrier in the spOHT model and identify the molecular players involved.

RESPONSE: Thank you for the suggestion. To evaluate the permeability of BRB in the spOHT model is beyond the scope of this study. We would assess the BRB permeability in future studies.